# Blue-Green (~480 nm) versus Blue (~460 nm) Light for Newborn Phototherapy—Safety Considerations

**DOI:** 10.3390/ijms24010461

**Published:** 2022-12-27

**Authors:** Finn Ebbesen, Hendrik Jan Vreman, Thor Willy Ruud Hansen

**Affiliations:** 1Department of Pediatrics, Aalborg University Hospital, 9000 Aalborg, Denmark; 2Department of Clinical Medicine, Aalborg University Hospital, 9000 Aalborg, Denmark; 3Division of Neonatal and Development Medicine, Department of Pediatrics, Stanford University Medical Center, Stanford, CA 94305-5101, USA; 4Division of Paediatric and Adolescent Medicine, Oslo University Hospital, 0424 Oslo, Norway; 5Faculty of Medicine, Institute of Clinical Medicine, University of Oslo, 0450 Oslo, Norway

**Keywords:** neonates, hyperbilirubinemia, phototherapy, side effects, safety, fluorescent light, LED light, blue-green light, blue light

## Abstract

We have previously shown that the phototherapy of hyperbilirubinemic neonates using blue-green LED light with a peak wavelength of ~478 nm is 31% more efficient for removing unconjugated bilirubin from circulation than blue LED light with a peak wavelength of ~452 nm. Based on these results, we recommended that the phototherapy of hyperbilirubinemic newborns be practiced with light of ~480 nm. Aim: Identify and discuss the most prominent potential changes that have been observed in the health effects of phototherapy using either blue fluorescent- or blue LED light and speculate on the expected effects of changing to blue-green LED light phototherapy. Search the phototherapy literature using the terms neonate, hyperbilirubinemia, and phototherapy in the PubMed and Embase databases. Transitioning from blue fluorescent light to blue-green LED light will expose neonates to less light in the 400–450 nm spectral range, potentially leading to less photo-oxidation and geno-/cytotoxicity, reduced risk of cancer, and decreased mortality in extremely low-birthweight neonates. The riboflavin level may decline, and the increased production and retention of bronze pigments may occur in predisposed neonates due to enhanced lumirubin formation. The production of pre-inflammatory cytokines may rise. Hemodynamic responses and transepidermal water loss are less likely to occur. The risk of hyperthermia may decrease with the use of blue-green LED light and the risk of hypothermia may increase. Parent–neonate attachment and breastfeeding will be positively affected because of the shortened duration of phototherapy. The latter may also lead to a significant reduction in the cost of phototherapy procedures as well as the hospitalization process.

## 1. Introduction

Currently, in Scandinavia, 2–3% of all late preterm and term (≥35 weeks) neonates and 40–80% of preterm neonates with lower gestational ages (<35 weeks) with hyperbilirubinemia need phototherapy [1,2]. The current standard involves the exposure of neonatal skin to blue light with a peak wavelength of ~460 nm. This light is absorbed by native Z,Z-bilirubin molecules which undergo conversion to stereoisomers, the configurational Z,E- and E,Z-bilirubin, and the structural isomers ZE- and E,E-lumirubin (Figure 1). These more polar isomers can be excreted with the bile and urine. Lumirubin formation is the most important process for reducing total serum bilirubin (TSB) because it is produced in large amounts and is excreted rapidly [3,4]. Z,Z-bilirubin also undergoes photo-oxidation with a very small quantum yield, forming, among others, monopyrrolic (BOX A and BOX B), dipyrrolic, and tripyrrolic oxidation products [5,6].

From the discovery of phototherapy in the 1950s to the present, broad bandwidth (~50 nm at 50% peak irradiance) blue fluorescent light with peak emission ~460 nm, matching the absorption spectrum of the bilirubin–albumin complex in plasma with peak absorption ~460 nm, has been used worldwide for phototherapy. However, fluorescent lamps emit low levels of ultraviolet light A (<400 nm), which is not always prevented from reaching the treated subjects. In addition, this light source also emits significant levels of infrared radiation (heat). In the early 2000s, narrow bandwidth (~20 nm at 50% peak irradiance) blue LED (light emitting diode) light with peak emission at ~460 nm began replacing the fluorescent lamps. During that period, using in vivo models of neonatal skin, studies of the action (efficacy) spectrum of phototherapy demonstrated that the optimum wavelength for phototherapy did not quite occur at ~460 nm, but rather at a wavelength some 20 nm higher [7,8,9,10]. Based on these observations, we conducted a series of clinical studies that ultimately demonstrated that blue-green (also identified as turquoise) LED light with a narrow bandwidth of 470–490 nm and a peak at 478 nm was 31% more efficient than blue LED light with a bandwidth of 450–470 nm and a peak at 459 nm [11]. Therefore, based on these clinical data, we recently recommended the use of this light source and emission characteristics [11,12], as had Lamola [10] and the American Academy of Pediatrics (AAP) [13].

The greater efficacy of blue-green LED light is maybe due to several factors: (1) When the light wavelength increases beyond 459 nm, the quantum yield of lumirubin increases, while the rate of the formation of Z,E-bilirubin decreases [14,15]. Moreover, (2) the competition between bilirubin and hemoglobin as well as melanin for light absorption decreases with wavelengths longer than 459 nm, while back-scattering is reduced [10]. We have shown clinically that the decline of TSB using blue LED light was negatively related to the hemoglobin concentration [11,16], while this was not the case for LED light with peak emission at 478 nm [11].

The most consequential side effect of phototherapy is considered to be photodynamic damage: When the bilirubin molecule absorbs a photon to yield isomerization, cyclization, and oxidation products, it can also act as a photosensitizer. The absorbed energy can be transferred to molecular oxygen to form singlet oxygen, superoxide di-anion, and its derivative, the hydroxyl radical. These products can subsequently react with water molecules to produce peroxides, such as hydrogen peroxides, and reactive oxygen species [17,18]. To prevent the formation of oxidants, as well as to repair oxidative damage, cells have an armory of enzymatic and non-enzymatic antioxidants, including superoxide dismutase, catalase, glutathione peroxidase, reduced glutathione, vitamins C and E, uric acid, etc. [17,18]. On the other hand, bilirubin has also been shown to act as an antioxidant that neutralizes tissue oxidants [19]. In fact, it has been suggested that physiologic TSB values are associated with antioxidant effects, while high pathologic values are associated with prooxidant effects [20].

The evaluation and interpretation of phototherapy studies present unique challenges, because so many parameters (device used; percent body surface area treated; type of light source; spectral quality of delivered light; irradiance level delivered; distance of lamp to skin; use of an appropriate spectrometer; duration of treatment; presence of ultraviolet and infrared wavelengths, etc.) all contribute to the magnitude of the lowering effect on total bilirubin in the circulation. However, many reports lack some or most of these essential parameters, which complicates and limits the interpretation of the reported data and the comparison of results between studies.

Herein, we aim to discuss the most likely changes in the side effects of phototherapy that might be observed if blue-green LED were to substitute blue fluorescent light sources for the treatment of jaundiced neonates.

## 2. Results

### 2.1. Immediate Side Effects

Photodynamic damage (photo-oxidation and geno-/cytotoxicity).

#### 2.1.1. In Vitro Studies

The oxidative effects in cells exposed to fluorescent phototherapy light in the presence of bilirubin have been studied in cell cultures [6,21,22,23,24,25,26]. An increased frequency of chromatid breaks and exchanges was seen only in cells exposed to blue light in the 400–450 nm spectral range [21]. Fewer single DNA strand breaks and longer cell survival were seen with blue-green fluorescent light in the 490–530 nm spectral range versus shorter wavelengths [22,23]. Roll et al. [24] found less damage, expressed as the inhibition of cell growth and necrosis, in cells exposed to blue-green fluorescent light with peak emission at 490 nm than in cells exposed to blue fluorescent light with peak emission at 450 nm. Thus, these in vitro studies appear to show that the exposure of cells to the higher wavelengths of blue-green light is associated with less photodynamic damage than blue light, leading us to speculate that avoiding the use of light in the 400–450 nm spectral range may lead to safer phototherapy for neonates.

When the bilirubin photoisomers Z,E-bilirubin, E,Z-bilirubin, and lumirubin, as well as COX A and COX B, were incorporated into growing media, none of these bilirubin products appeared to exert any toxic effects on cell viability [6,25]. Lumirubin was found to induce fewer changes in mitochondrial respiration, substrate metabolism, and reactive oxygen species than bilirubin IXα Z,Z [26].

#### 2.1.2. Animal Studies

When hyperbilirubinemic homozygous Gunn rats were exposed to 440–520 nm blue-green LED light [27,28], they excreted considerably less 8-hydroxydesoxyguanosin (8-OHdG) in the urine than rats exposed to light in the wider spectral range 400 to 520 nm [27]. Furthermore, Gunn rats exposed to 450–520 nm light excreted the same amount of 8-OHdG as non-exposed animals [28]. 8-OHdG is an oxidation product of guanine, which is the most light-sensitive base of DNA [29]. This suggests that the exclusion of low wavelength light in the 400–450 nm range, in favor of that in the 450–500 nm range, reduces photodynamic DNA damage [27,28]. However, Gunn rats that received intensive phototherapy in the form of LED light with a peak emission of 467 nm (range 416–544 nm) and irradiance of 100 µW/cm^2^/nm had normal excretion of 8-OHdG and normal content of gamma-H2AX (a phosphorylated histone protein)-positive cells in the skin, a marker of DNA repair [28]. Because bilirubin metabolism is somewhat different in humans than in Gunn rats due to different properties of the albumin molecule [30], the implications of these results for humans need to be ascertained. Moreover, contrary to humans, the production and excretion of lumirubin in rats are low compared to ZE- and EZ-bilirubin [31].

#### 2.1.3. Neonatal Studies

Oxidant stress and antioxidant defenses were compared between neonates exposed to either broad-spectrum blue fluorescent light or narrow-spectrum blue LED light with peaks of ~460 nm [32,33,34,35,36]. Thus, with LED light, the neonates are exposed to less light in the 400–450 nm range. Total oxidant substances (TOS) increased, total antioxidant capacity (TAC) decreased, and the oxidant stress index (OSI = TOS/TAC) increased in all groups, while the frequency of sister chromatid exchange (SCE) and DNA damage scores increased; however, the observed changes differed between the light sources [32,33,34,35,36]. In summary, the described changes in OSI were greatest in neonates exposed to blue fluorescent light [33,35] or blue LED light [32] or did not differ significantly between light sources [34]. In Mohamed et al.’s study [32] the frequency of SCE and DNA damage score increased most in neonates receiving LED light, while Karadag et al. [36] failed to find differences in SCE between groups that received LED phototherapy versus fluorescent phototherapy when compared to jaundiced controls. SCE frequency and DNA damage correlated positively with TOS [32]. Thus, oxidant stress seems to increase in all groups of jaundiced neonates who receive fluorescent or LED light, and there seems to be no significant difference between the two light sources with respect to these parameters. However, because irradiance levels were different, conclusions must be cautious: thus, the irradiance levels in neonates exposed to fluorescent light were 10–20 µW/cm^2^/nm compared to 30–35 µWatt/cm^2^/nm in neonates exposed to LED light. Blue LED light with peak emission ~460 nm and irradiances up to 35 µW/cm^2^/nm did not induce oxidative DNA damage, expressed by the urine excretion of 8-OHdG, in neonates with gestational age ≤ 32 weeks [37].

### 2.2. Extremely Low Birth Neonates

In particular, during the phototherapy of extremely low-birthweight neonates (ELBW), potential side effects need to be minimized. In a monumental study [38,39,40], one group of ELBW neonates was treated “aggressively”, while another group was treated “conservatively”, i.e., in the aggressively treated group phototherapy was initiated at lower TSB concentrations (either >85 µmol/L (5.0 mg/100 mL) or >119 µmol/L (7.0 mg/100 mL) depending on the body weight range) than in conservatively treated group (either >135 µmol/L (7.9 mg/100 mL) or >171 µmol/L (10.0 mg/100 mL)). The irradiances were comparable. As might be expected, the aggressively treated group had significantly lower TSB after the treatment and, for a given TSB level, a longer duration of treatment than the conservatively treated group. Follow-up at 18 to 22 months of age showed that aggressively treated neonates with a birth body weight range of 500 to 750 g, who had respiratory failure in need of mechanical ventilation, had a 5% reduction in the rates of neurodevelopment impairment, but also a 5% higher mortality. This higher mortality could have been due to photodynamic damage in these most immature babies with (1) limited photoprotection capacity due to very thin, translucent skin with significantly reduced stratum corneum, low melanin content in the immature melanocytes [41], and a low blood hemoglobin concentration, which might compete with bilirubin for the light [16], and (2) low antioxidant capacity [41]. Another explanation might be a longer period of hemodynamic instability accompanying phototherapy [42]. The neonates were treated almost equally with either blue fluorescent light, a halogen spotlight, or blue LED light. The irradiances were maintained between 15 and 40 µW/cm^2^/nm [43]. This means that the risk in mortality of future ELBW neonates might decline by treating all neonates with blue-green LED light due to potentially reduced photodynamic damage and more hemodynamic stability (see Section 2.6). However, using longer wavelengths, the formation of lumirubin increases, and in vitro studies have shown that lumirubin can affect the differentiation of human pluripotent cell-derived stem cells, i.e., it might affect early neurodevelopment [44].

### 2.3. Riboflavin Level

Riboflavin is highly sensitive to light. The absorption maximum of the riboflavin molecule in vitro is at 447 nm, close to the absorption maximum of the bilirubin–albumin complex [45] and close to the peak emission wavelengths of blue fluorescent and LED light at ~460 nm. However, the peak of the action spectrum of riboflavin in neonates may be altered due to the back-scattering of light and competing photon absorption by oxyhemoglobin, desoxyhemoglobin, and melanin, as has been determined for bilirubin [10]. By the absorption of a photon, riboflavin in the skin can act as a photosensitizer and transfer energy to molecular oxygen, yielding hydrogen peroxide and oxidative products of riboflavin itself [46], thereby reducing the riboflavin level [45,47,48]. Using blue-green LED light instead of blue light, we hypothesize that the decomposition rate of riboflavin will most likely be increased.

Studies in vitro as well as in Gunn rats showed that riboflavin also enhances the photo-dynamic destruction of bilirubin, resulting in decreased TSB [49,50,51]. Presumably, this will also occur in neonates [49,52]. Thus, a change in phototherapy light quality to longer wavelengths may also cause a change in the rate of bilirubin alteration via this mechanism.

### 2.4. Bronze Baby Syndrome

The bronze baby syndrome is a rare, visible side effect of phototherapy strongly associated with liver diseases. The atresia of extra- or intrahepatic bile ducts, severe hemolysis especially due to Rhesus and AB0 blood types isohemolytic diseases and severe bacterial sepsis are predisposing factors [53,54,55,56]. The syndrome was thought to be caused by the retention of copper–porphyrin complexes changed to bronze pigments by the photosensitization of bilirubin [54]. However, this hypothesis has been shown to be unlikely [57]. Ito et al. [42] suggest that the syndrome is caused by the accumulation of polymerized products of bilirubin photoisomers, predominantly lumirubin. If this hypothesis is correct, the use of LED light with longer wavelengths than 460 nm could, by increasing the production of lumirubin, presumably also increase the production and retention of bronze pigments in neonates with predisposing factors, if this compound is not excreted rapidly. Although it is generally believed that the syndrome is harmless, there are some suggestions that it may constitute an additional risk of KSD [55,56].

### 2.5. Immune System

Phototherapy can affect the immature immune system via the alteration of cytokine production (Table 1) [25,58,59,60]. The exposure of hyperbilirubinemic neonates to a mixture of blue and white fluorescent light was found to increase the plasma concentrations of the pro-inflammatory cytokines TNF-α, IL-1ß, and IL-8, and reduce the lymphocyte subset CD3+, while the concentration of IL-6 remained unchanged [58]. Using the same types of light the phototherapy resulted in a significant decrease in IL-6, but without significant changes in TNF-α, IL-1ß, IL-8, and IL-10 [59]. An increase in irradiance was associated with a greater decline in IL-6 [59]. Furthermore, during phototherapy, the production of IL-1ß by peripheral blood monocytes decreased, and IL-2 and the anti-inflammatory cytokine IL-10 increased [60]. Jasprova et al. [6] showed that lumirubin upregulated the gene expression of TNF-α, IL-1ß, and IL-6 in rat hippocampal slices. However, immunologic studies comparing different light qualities have not been performed, and it will be apparent from the results cited above that the reported impact of phototherapy on immunological processes varies considerably, presumably due to methodological differences, including wavelength determinations and irradiance measurements. Thus, further studies are needed to elucidate these phenomena. As an example, blue-green LED light increases the production of lumirubin [3,4], which, according to the findings of Jasprova et al. [6], may perhaps induce a greater immune response.

### 2.6. Hemodynamic Changes

The exposure of preterm neonates to blue fluorescent light has been shown to change the hemodynamics in various organs. Peripheral blood flow increased due to the relaxation of the blood vessels in the skin [61,62], presumably caused by visual light with wavelengths above 530 nm and/or infrared light (heat) [63]. This was followed by the redistribution of the blood flow: cerebral blood flow increased, while renal, as well as postprandial mesenteric blood flow and cardiac output, decreased [64,65,66]. A small but significant drop in blood pressure and an increase in heart rate occurred in low-birthweight neonates [66]. The incidence of open ductus arteriosus was increased in ELBW neonates, thus a reopening of the duct may occur [66]. The dilatation of the skin blood vessels may result from a changed dynamic balance between the vasodilator nitric oxide and the constrictor endothelin, in the form of an increased ratio of nitric oxide to endothelin, the plasma concentrations of which both rise during phototherapy with fluorescent light [67]. However, cerebral blood flow did not increase in thermally stable preterm neonates exposed to blue fluorescent light (care was taken to avoid heat stress caused by irradiation from the phototherapy equipment) [68]. In neonates exposed to blue LED light, changes in cerebral and mesenteric blood flows were not seen, most likely because this light source emits very little light with wavelengths > 530 nm, including infrared light [63]. Most likely, similar results will be seen in response to treatment with blue-green LED light.

### 2.7. Transepidermal Water Loss

In neonates treated with daylight or blue fluorescent light, a rise in transepidermal water loss (TEWL) was seen [63,69,70], presumably caused by light with wavelengths > 530 nm, including infrared light [63], while in thermally stable term neonates, TELW did not change, as expected [69]. In preterm neonates receiving blue LED phototherapy of standard irradiance, TELW did not change, most likely because this light source emits very little heat in the above-mentioned spectral range [63]. Therefore, such neonates do not need extra fluid during phototherapy. It is very likely that similar results will be obtained using blue-green LED light.

### 2.8. Hyper- and Hypothermia

Blue fluorescent phototherapy increased the body temperature in both preterm and term neonates, while blue LED phototherapy with an irradiance < 60 µW/cm^2^/nm did not cause a change in body temperature. However, at irradiances > 60 µW/cm^2^/nm, a risk of hyperthermia was observed, even with the use of LED light [71]. Blue LED phototherapy involved a low risk of hyperthermia because LEDs emit little heat [72]. However, depending on the distance of the body to the light source, neonates, especially premature ones, may be at risk for hypothermia. Thus, it is recommended that the body temperature also be monitored during LED phototherapy, especially when the neonate is treated in a bassinet. It is most likely that body temperature will behave similarly with the use of blue-green LED phototherapy.

### 2.9. Vision

It is deemed unlikely that a switch in use from short wavelength blue to longer wavelength blue-green light will cause eye damage, as there are no clinical reports that the shorter blue wavelength causes any measurable eye damage. However, the eyes of hospitalized neonates are routinely protected by eye patches when under phototherapy. The evidence for this practice has been derived from a limited number of controversial rat studies [73,74]. When Crigler-Najjar patients, who typically receive high irradiance levels of phototherapy (up to 100 µW/cm^2^/nm) over large proportions (>30%) of their body surfaces for 8–10 h/day for many years, were tested for visual acuity and color discrimination score, no significant difference in these qualities were found when compared to age-matched non-Crigler-Najjar sibling controls. The Crigler-Najjar subjects typically do not wear eye protection while receiving phototherapy [75,76].

### 2.10. Hypocalcemia

During phototherapy with blue or white fluorescent light, a usually asymptomatic hypocalcemia was found [77,78,79,80]. Plasma levels usually returned to normal by 24 h post-phototherapy [77]. It was hypothesized that the hypocalcemia was due to a decrease in pineal melatonin secretion, induced by transcranial penetration of the light [81]. By Covering the head, the decline of serum calcium was reduced, but it was still significant [80]. As neonates wear eye pads during phototherapy, the hypocalcemic effect may involve extra-ocular pathways. It was suggested that melatonin may block the absorbing effect of cortisol on bone calcium [81]. Finally, the decline in serum calcium during phototherapy was neither correlated to TSB [78] nor plasma melatonin concentrations [79]. More studies are needed to show whether the risk of hypocalcemia will change with the use of blue-green LED phototherapy.

### 2.11. Loose Stools

During phototherapy with blue fluorescent light, loose green stool is frequently seen, and the gut transit time is decreased in a dose-dependent manner [82]. The increased intestinal secretion of water and electrolytes during phototherapy may be induced by the increased production of vasoactive intestinal peptides, possibly caused by bilirubin isomers or bilirubin degradation products [83]. Future investigations may show whether a change to blue-green LED light will change the gut transit time.

### 2.12. Parent–Neonate Attachment and Breastfeeding

Parent–neonate separation during phototherapy from above and/or below has been reported to have a negative influence on the establishment of parent–neonate social interaction and breastfeeding, as well as making nursing access and observations more cumbersome. In addition, a number of families and caregivers have reported issues when exposed to blue and blue-green fluorescent light (14). It is likely that these consequences will be reduced by a switch from blue- to blue-green LED phototherapy due to the shorter duration of phototherapy. However, the caregivers’ tolerance of blue and blue-green fluorescent light was equal (14).

## 3. Long-Term Side Effects (Epidemiologic Studies)

### 3.1. Cancer

Among the long-term side effects of phototherapy, the risk of cancer is potentially the most serious and often-discussed topic. As mentioned above, the phototherapy of neonates may damage DNA, increase oxidant stress, and produce cytokines, factors that have all been implicated in the pathogenesis of cancer [84]. Thus, phototherapy has carcinogenic potential [85].

Newman et al. [84] found a slightly positive association between phototherapy and any leukemia, non-lymphocytic cancer, and liver cancer, but after adjustment for confounders, including TSB, these associations were eliminated or diminished and were no longer significant. Both for neonates who either received or did not receive phototherapy, in adjusted analyses significant positive associations were found between hyperbilirubinemia and infantile myeloid leukemia and kidney cancer [86], any childhood leukemia [84], childhood myeloid leukemia [87], and solid tumors, especially CNS and nervous system tumors [85]. TSB below the phototherapy limit was also associated with cancer development [84,86], but the associations were stronger for TSB values above the phototherapy limit in neonates in need of phototherapy [85,86,87,88]. These associations may either simply reflect groups of infants exposed to more severe hyperbilirubinemia or may reflect added cancer risk due to phototherapy. In other studies, no association was found between neonatal hyperbilirubinemia and leukemia in neonates, who either received or did not receive phototherapy [89,90]. To our knowledge, no cases of cancer have been reported in patients suffering from Crigler-Najjar syndrome. In summary, based on the current state of knowledge, we cannot rule out a possible association between phototherapy and the risk of tumor development, but the evidence for such a side effect is very weak, especially as the vast majority of these neonates have, in all probability, been exposed to blue fluorescent light. We speculate that, if neonates are treated with blue-green LED light, both DNA damage and oxidative stress will decrease, and possibly also the cancer risk.

### 3.2. Allergic Diseases

An association between the incidence of allergic diseases (asthma, allergic conjunctivitis and rhinitis, atopic dermatitis, and urticaria) and neonatal hyperbilirubinemia has been observed in several studies, both in children who either received or did not receive phototherapy [91,92,93]. These associations may be enhanced by phototherapy, as it increases the production of Th-2 pro-inflammatory cytokines such as TNF-α, IL-1ß, and IL-8 [58], and also decreases IL-6 [59], which promotes a Th-2 shift in a pro-allergic direction [93,94]. Because the changes in interleukin concentrations may be induced by lumirubin [23], the association may become stronger using blue-green LED light than blue LED light due to the increased lumirubin production [14,15].

Eosinophils have a role in the pathogenesis of childhood asthma. After the phototherapy of jaundiced neonates using blue LED light, the blood levels of eosinophils and eosinophilic cationic protein increased as an expression of the activation of eosinophils [95,96,97].

### 3.3. Diabetes Type 1

A small but significant association between diabetes mellitus type 1 and neonatal hyperbilirubinemia, with or without the need for phototherapy, has been described [98,99]. However, Newman et al. [100] could not confirm this association.

### 3.4. Childhood Epilepsy

Hyperbilirubinemia was associated with an increased risk for childhood epilepsy, primarily in boys, both in children who had received versus had not received phototherapy [101,102]. These investigations confirmed earlier studies showing a positive association between neonatal hyperbilirubinemia and childhood epilepsy [103,104].

### 3.5. Autism Spectrum Disorders

Studies on the association between autism spectrum disorders, hyperbilirubinemia, and phototherapy have shown divergent results: both a positive relationship [105,106] or a lack of a relationship have been demonstrated [107,108,109]. These inconsistent results may be due to methodological differences.

## 4. Summary

An overview of the potential immediate and long-term side effects resulting from a switch from blue fluorescent light to blue-green LED light is shown in Table 2.

In the epidemiological studies of diabetes type 1, childhood epilepsy, and autism spectrum disorders, the types of light sources were not reported, but the vast majority of children have, in all probability, been exposed to blue fluorescent light. Whether these potentially weak side effects will be confirmed or even more diminished with the use of blue-green LED light needs to be determined with future investigations.

## 5. Material and Methods

A search of the literature with the terms neonate, hyperbilirubinemia, and phototherapy in PubMed and Embase.

## 6. Conclusions

Transitioning from the traditional blue fluorescent light to blue-green LED phototherapy will expose neonates to less light in the 400–450 nm spectral range, hypothetically leading to less photodynamic damage, reduced risk of cancer, and decreased mortality in ELBW neonates. The degree of riboflavin deficiency may be greater, and the increased production and retention of bronze pigments may possibly occur in predisposed neonates due to the enhanced formation of lumirubin. The production of pre-inflammatory cytokines may increase. Changes in hemodynamic responses and TEWL may not occur. The risk of hypothermia is expected to increase during blue-green phototherapy, while the risk of hyperthermia is expected to decrease. The negative influence on parent–neonate attachment and breastfeeding may be reduced due to the shorter duration of the phototherapy. The risk of allergic diseases may increase due to a greater immune response.

Currently, we cannot predict with reasonable certainty whether the possible risks of immediate side effects such as hypocalcemia and loose stool, as well as long-term side effects such as diabetes type 1, epilepsy, and autism spectrum disorders, will change upon transitioning from blue to blue-green LED light. This will need to be explored in future investigations. However, we wish to emphasize that the major benefit of shifting from blue to blue-green phototherapy is the very significant increase in efficacy (30%) leading to **a**: a reduction in time under phototherapy; **b**: a shorter exposure of the body to phototherapy light and its consequences; **c**: a shorter exposure to elevated, unconjugated bilirubin concentrations; **d**: reductions in the costs of phototherapy; and finally **e**: a reduction in the cost of phototherapy involving hospitalizations.

Finally, although some associations were shown in the epidemiologic studies, this does not constitute proof of a cause–effect relationship. Indeed, they may be epiphenomena.

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
