# Peer review of "Blue-Green (~480 nm) versus Blue (~460 nm) Light for Newborn Phototherapy—Safety Considerations"

_ijms, 2022, doi:10.3390/ijms24010461_

Round 1

Reviewer 1 Report

In this work, Ebbesen and colleagues reviewed the safety considerations on the use of blue-green light vs blue light for phototherapy in newborns.  They consider that the advantages outweigh the disadvantages, even though more studies are needed.  I find the manuscript to be well written and should be ready for publishing after the authors address some minor issues and suggestions:

1. Consider inserting the chemical reactions described in the manuscript and how light catalyzes it.

2. Please check for misspellings and writing mistakes.  For example, in the second paragraph of page 2 “…The (…) efficacy of (…)LED light is may due to…” would be better as ““…The (…) efficacy of (…)LED light is maybe due to…”.

3. Please, consider referencing the properties of the light source used in the study discussed in the first phrase of section 1.2.

4. In section 1.4., consider inserting the predisposing factors discussed.

5. For studies 58 and 59 I missed to understand the differences between the light sources used.  Please, consider addressing this.

Author Response

Dear reviewer

Thank you for a comprehensive review

Reviewer 1.

  1. Consider inserting the chemical reactions described in the manuscript and how light catalyzes it.

Response: Figure 1 is inserted showing the chemical reactions (page 2, line 4 from beneath).

Figure 1. Formation of bilirubin isomers during phototherapy of hyperbilirubinemic neonates.

                                                Light                                                                                                 

                                                    á´—

 E,Z-bilirubin        ↔        Z,Z-bilirubin        ↔        Z,E-bilirubin

        ↓ 

E,Z-lumirubin      ↔        E,E-lumirubin 

  1. Please check the misspellings and writing mistakes.

Response: “may” is replaced by “maybe” (p. 2, l. 21).

  1. Please, consider referencing the properties of the light source used in the study discussed in the first phrase of section 1.2.

Response: “The irradiances were maintained between 15- and 40 µW/cm2/cm” are inserted (p.4, l. 30).

  1. In section 1.4 consider inserting the predisposing factors discussed.

Response: “Obstructions of the bile ducts, liver diseases, severe hemolysis and sepsis” are replaced by: “Atresia of the extra- or intrahepatic bile ducts, severe hemolysis most often blood types Rhesus- or AB0 isohemolytic diseases and severe bacterial sepsis [53 – 56]” (p. 5, l. 5-6).

Further, the numbers of the references are changed: Previous reference 55 is now called reference 57, and previous references 56 and 57 are called references 55 and 56.

  1. For studies 58 and 59 I missed to understand the differences between the light sourses used. Please, consider to addressing this.

Response: “During similar clinical conditions” is replaced by “Using the same sorts of light” (p.5 l. 21-22). 

Reviewer 2 Report

It is a pleasure to review this manuscript. Presented studies are clear and well reported. The manuscript provides evidence-based data about phototherapy, which is quite standardized and safe for term and near-term newborns, but potentially harmful for ELBW infants. Indeed, it is common in clinical practice a reduced tolerance to enteral feeding during phototherapy (33906567). Thus, it would be helpful to reduce the duration of phototherapy among these infants with blu-green LED phototerapy. However, conclusions are too firm and not supported by enough evidence. In conclusion, the manuscript merits publication with major revisions.

-    1. It appears to me that the article type is a review.

-     2.    Page 4: Follow-up at 18 to 22 months of age showed that aggressively treated
neonates with birth body weight range of 500 to 750 g, who had respiratory failure in need
of mechanical ventilation, had a 5% reduction in neurodevelopment, but also a 5% higher
mortality

o   What does it mean a “reduction in neurodevelopment”? Do you mean worse neurodevelopment? Please clarify

-    3.     Page 4: “Another explanation might be a longer period of hemodynamic instability accompanying phototherapy [42]. The neonates were treated almost equally with either blue fluorescent light, halogen
spotlight, or blue LED light [43].

o   How phototherapy can affect hemodynamic stability of ELBW?

o   On what basis do you predict that blu-green LED light may improve hemodynamic stability?

-      4.    Conclusion: “potentially leading to less photodynamic damage, reduced risk of cancer, and decreased mortality in ELBW neonates.

o   The evidence supporting these potential benefits appear weak. Indeed, the risk of cancer is speculative and the higher hemodynamic instability and its related increased mortality in ELBW infants exposed to phototherapy is a hypothesis.

-           

Author Response

Dear reviewer

Thank you for a comprehensive review.

  1. It appears to me that the article type is a review.

                     Response: We have discussed the article type intensively, and we do not mean, it is a review.

  1. Page 4. Follow-up at 18 – 24 months of age showed that aggressively treated neonates with birth body weight range of 500 g to 750 g , who had respiratory failure in need of mechanical ventilation had a 5% in reduction of neurodevelopment , but also a 5% higher mortality.

                     What does it mean a “reduction in neurodevelopment”?. Do you mean worse      

                     neurodevelopment? Please, clarify.

                     Response: “5% reduction in neurodevelopment” is replaced by “5% reduction in the rates of  

                      neurodevelopment impairment? (p. 4 l. 22).

  1. Page 4: “Another explanation might be a longer period of hemodynamic instability

                      accompanying phototherapy [42]. The neonates treated almost equally with either blue        

                      fluorescent light, halogen spotlight, or blue LED light [43]. How phototherapy can affect       

                      hemodynamic stability of ELBW?

                      On what basis do you predict that blue-green LED light may improve hemodynamic stability?

                      Response: These questions are answered in paragraph 1.6. Hemodynamic changes. “See

                      paragraph 1.6” is inserted (p. 4 l. 32-33).

  1. “potentially leading to less photodynamic damage, reduced risk of cancer, and

         decreased mortality in ELBW neonates”.

                       The evidence supporting these potential benefits appear weak. Indeed, the risk of cancer is          

                        speculative and the higher hemodynamic instability and its related increased mortality in     

                        ELBW infants exposed to phototherapy is a hypothesis.

                       Response: Yes, we need to perform a weakness of the conclusion. Therefore “…… “potentially      

                       leading to less photodynamic damage” is replaced by “hypothetically leading to less    

                      photodynamic damage” (p. 9 l. 6 beneath Table 2). 

Round 2

Reviewer 2 Report

The authors have adequately addressed reviewers comments.